

# Vascular synovial phenotype indicates poor response to JAK inhibitors in rheumatoid arthritis patients: a pilot study

Mengxia Liu*, Pengcheng Liu*, JianBin Li, Yiping Huang and Rui Wu

Department of Rheumatology, The First Affiliated Hospital, Jiangxi Medical College, Nanchang University, Nanchang, Jiangxi Province, China
* These authors contributed equally to this work.

## ABSTRACT

**Objective:** Rheumatoid arthritis (RA) is a chronic autoimmune inflammatory disease, characterized by significant individual variations in treatment response. Predicting treatment response remains a formidable challenge. This study aims to identify predictors within the synovium associated with the response to JAK inhibitor therapy in RA patients, employing a retrospective approach.

**Methods:** A retrospective analysis was conducted on 27 RA patients who underwent synovial biopsy and received JAK inhibitor therapy for at least three months at our center, from January to November 2023. These patients had comprehensive clinical records. Based on their response to JAK inhibitor therapy, as measured by the ACR20 criteria, they were categorized into non-responder and responder groups. We compared clinical data (including sex, age, disease duration), laboratory findings (RF, ACPA, ESR, CRP, *etc.*), DAS28-CRP scores, and synovial pathology features—such as synovial lining hyperplasia, neovascularization, stromal activation, inflammatory infiltration, and the expression of CD3, CD20, CD68, and CD138 markers in the synovium—between the two groups.

**Results:** The rate of non-responder to JAKi was found to be 33.3% (nine cases out of a total of 27 patients). Non-responders, when compared to responders, exhibited longer disease duration, more pronounced synovial neovascularization alongside reduced infiltration of labeled CD20+ and CD138+ cells in the synovium. Multivariate regression analysis identified synovial neovascularization and disease duration as independent predictors of a poor response to JAK inhibitor treatment.

**Conclusions:** The presence of vascular phenotype with low inflammation within the synovium of RA patients is an indicator of poor response to JAK inhibitor therapy, highlighting its potential as a predictive marker for treatment outcomes.

# INTRODUCTION

Rheumatoid arthritis (RA) is a chronic autoimmune disorder primarily affecting the joints, characterized by synovitis and joint damage (*Lee & Weinblatt, 2001*). The implementation of the treat-to-target strategy in RA has demonstrated improved clinical outcomes, including enhanced symptom control, preservation of joint function, and a reduction in

Corresponding author
Rui Wu, ndyfy00400@ncu.edu.cn

radiographic joint damage (*Zink & Albrecht, 2017*; *Quinn & Emery, 2003*). Treatment options encompass nonsteroidal anti-inflammatory drugs (NSAIDs), glucocorticoids, disease-modifying antirheumatic drugs (DMARDs), and biologics (*Alivernini, Firestein & McInnes, 2022*). Recently, Janus kinase inhibitors (JAKi) have emerged as a widely used therapy for RA. When traditional DMARDs like methotrexate are ineffective or poorly tolerated in RA patients, JAKi therapy is recommended due to its significant efficacy in reducing inflammation and slowing down joint damage (*Fraenkel et al., 2021*).

JAKi, targeting the Janus kinase enzymes involved in the inflammatory process, have played a crucial role in the treat-to-target strategy for RA (*Aletaha & Smolen, 2018*). Clinical trials and real-world studies consistently indicate that JAKi, whether used as monotherapy or in combination with traditional DMARDs like methotrexate, can significantly reduce disease activity and improve patient outcomes (*Taylor et al., 2017*). Compared to biologics, JAKi offer the advantages of oral administration and lower cost, thus gaining wider application in RA patients with poor response to methotrexate (*Liu et al., 2022*). However, not all patients achieve the expected results, with approximately 30% of RA patients exhibiting an inadequate response to JAKi (*Fleischmann et al., 2012*).

Early prediction of response to JAKi treatment not only aids in minimizing unnecessary healthcare costs but also enables more precise treatment planning, thereby alleviating patient suffering and enhancing prognosis. However, forecasting the response to JAKi therapy remains a complex and evolving endeavor, with no single predictor or method ensuring accuracy for all cases (*Zhang et al., 2014*; *Zeng et al., 2021*). Despite ongoing research endeavors and the exploration of potential biomarkers, attaining precise prediction of treatment outcomes with JAKi continues to pose a challenge in the medical domain.

The central pathology of RA is synovitis, characterized by significant alterations in the synovium, such as synovial lining hyperplasia, neovascularization, stromal activation, and immune cell infiltration. However, the extent and nature of these changes vary among individuals. Previous research utilizing a combination of global gene expression, histologic, and cellular analyses has identified four distinct synovial phenotypes in RA patients: lymphoid, myeloid, low inflammatory, and fibroid, each displaying unique underlying gene expression patterns (*Kay & Upchurch, 2012*). These studies suggest that different synovial phenotypes may respond differently to treatment. However, there is currently limited information available regarding synovial phenotypes associated with response to JAKi.

We initiated a study in January 2023 involving 27 active RA patients who had received at least 3 months of JAK inhibitor therapy. The study aims to identify synovial predictors associated with the response to JAK inhibitor therapy in RA patients.

## PATIENTS AND METHODS

### Patients and study design

In this retrospective study, we collected data from 27 RA patients who underwent synovial biopsies and were treated with JAKi for at least three months. The research was conducted at the Department of Rheumatology and Immunology of the First Affiliated Hospital of

Nanchang University from January to November 2023. Participant eligibility was based on the 2010 American College of Rheumatology/European League Against Rheumatism (ACR/EULAR) diagnostic criteria for RA. Written informed consent was obtained from all participants before the synovial biopsy, adhering to the ethical standards upheld by the institutional review board. The ethical review board of the First Affiliated Hospital of Nanchang University granted ethical approval for this study under the ethics number: IIT [2023] Clinical Ethics Review No. 011.

We diligently gathered clinical data, encompassing disease activity evaluations, pain assessments, and joint function measurements. Laboratory tests were also performed to determine levels of inflammatory markers, such as C-reactive protein (CRP) and erythrocyte sedimentation rate (ESR), and autoantibodies, including rheumatoid factor (RF) and anti-citrullinated protein antibody (ACPA). Patients received regular monitoring throughout their JAKi treatment. Follow-up evaluations, conducted three months after initiating treatment, included both clinical examinations and laboratory tests to assess disease activity and evaluate treatment responses.

Disease activity was measured using the Disease Activity Score in 28 joints based on CRP (DAS28-CRP), which is calculated as follows: DAS28-CRP = [0.56 * sqrt (Tender Joint Count)] + [0.28 * sqrt (Swollen Joint Count)] + [0.36 * ln (CRP + 1)] + (0.014 * Patient's Global Assessment of Disease Activity). The assessment of treatment response adhered to the American College of Rheumatology (ACR) response criteria, categorizing patients into non-responder or responder groups based on the achievement of ACR20 criteria.

## Synovial biopsy and tissue processing

Synovial tissue samples were procured from inflamed joints employing a novel biopsy needle, as illustrated in Fig. 1. The biopsy procedure is depicted in Fig. 2, showcasing the methodical approach to obtaining synovial tissue. Post-collection, the tissue was fixed in 4% paraformaldehyde, buffered with PBS, for a duration of 48 h. This was followed by a process of dehydration and subsequent embedding in paraffin. The paraffin-embedded tissue blocks were then sectioned into slices 3–4 μm in thickness, which were stained with hematoxylin and eosin (H & E) for microscopic examination. This process allowed for the detailed assessment of synovial structure under a light microscope. The histopathological evaluation of the synovial tissue focused on key features such as synovial lining hyperplasia, stromal activation, neovascularization, and the extent of inflammatory infiltration. Each of these aspects was semi-quantitatively scored on a scale from 0 to 3, providing a structured assessment of the tissue pathology.

Furthermore, immunohistochemistry techniques were applied to identify immune cells expressing specific lineage markers, including CD20, CD3, CD68, and CD138. Cells stained positive for these markers were meticulously observed under a 10× high-power microscope. Areas of high cell density were pinpointed for accurate cell counting, with the count of positively marked cells per 20× high-power field in the sublining layer meticulously recorded for each marker, facilitating a comprehensive analysis of immune cell infiltration.

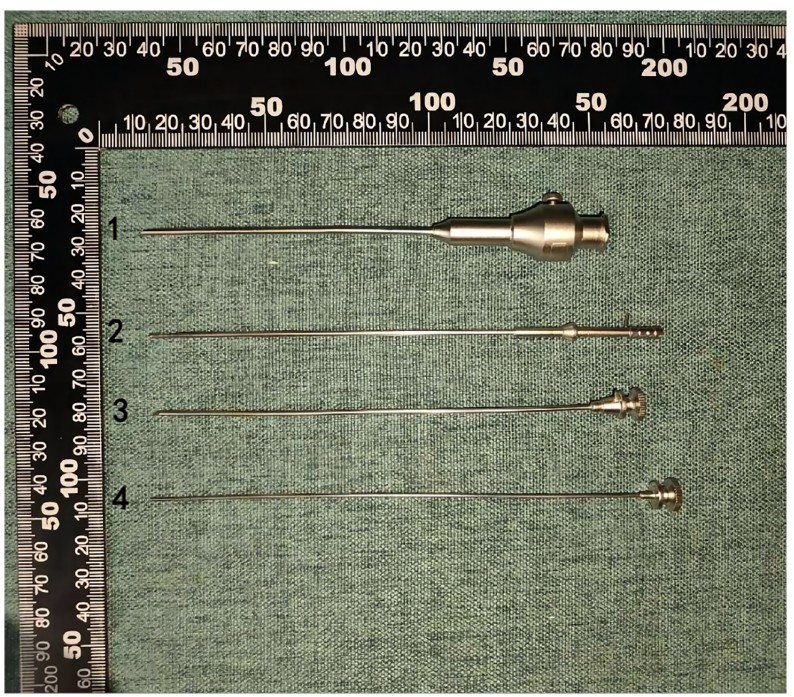

**Figure 1 The structure of the new synovial biopsy device.** The device primarily consists of a puncture sheath needle (1), a puncture inner core (2), a sampling needle (3), and a sampling needle core (4).

## Data analysis

We utilized SPSS version 26.0 (IBM, Armonk, NY, USA) for statistical analysis. The Shapiro-Wilk (SW) test was used to assess the normality of quantitative data. For normally distributed data, the mean ± standard deviation was reported, and group comparisons were performed using an independent sample t-test. For quantitative data that were not normally distributed, the median along with the 25th percentile and the 75th percentile were used for description, and the Mann-Whitney U test was employed for comparisons between groups. Qualitative data were expressed as frequencies and percentages, and group comparisons were conducted using Fisher's exact test. To investigate the adverse factors associated with JAK inhibitor response, binary logistic regression analysis was performed. All statistical tests were two-sided, and a $P$-value of less than 0.05 was considered statistically significant.

## RESULTS

### Baseline clinical characteristics of all patients in two groups

A total of 27 patients with RA were enrolled in the study, with three receiving baricitinib treatment and 24 receiving tofacitinib. Among these patients, 9 (33.3%) failed to achieve ACR20 response and were classified into the non-responder group, while the remaining patients were included in the responder group.

Baseline clinical characteristics are summarized in Table 1. No significant differences were observed in major clinical characteristics between the two groups. Disease duration

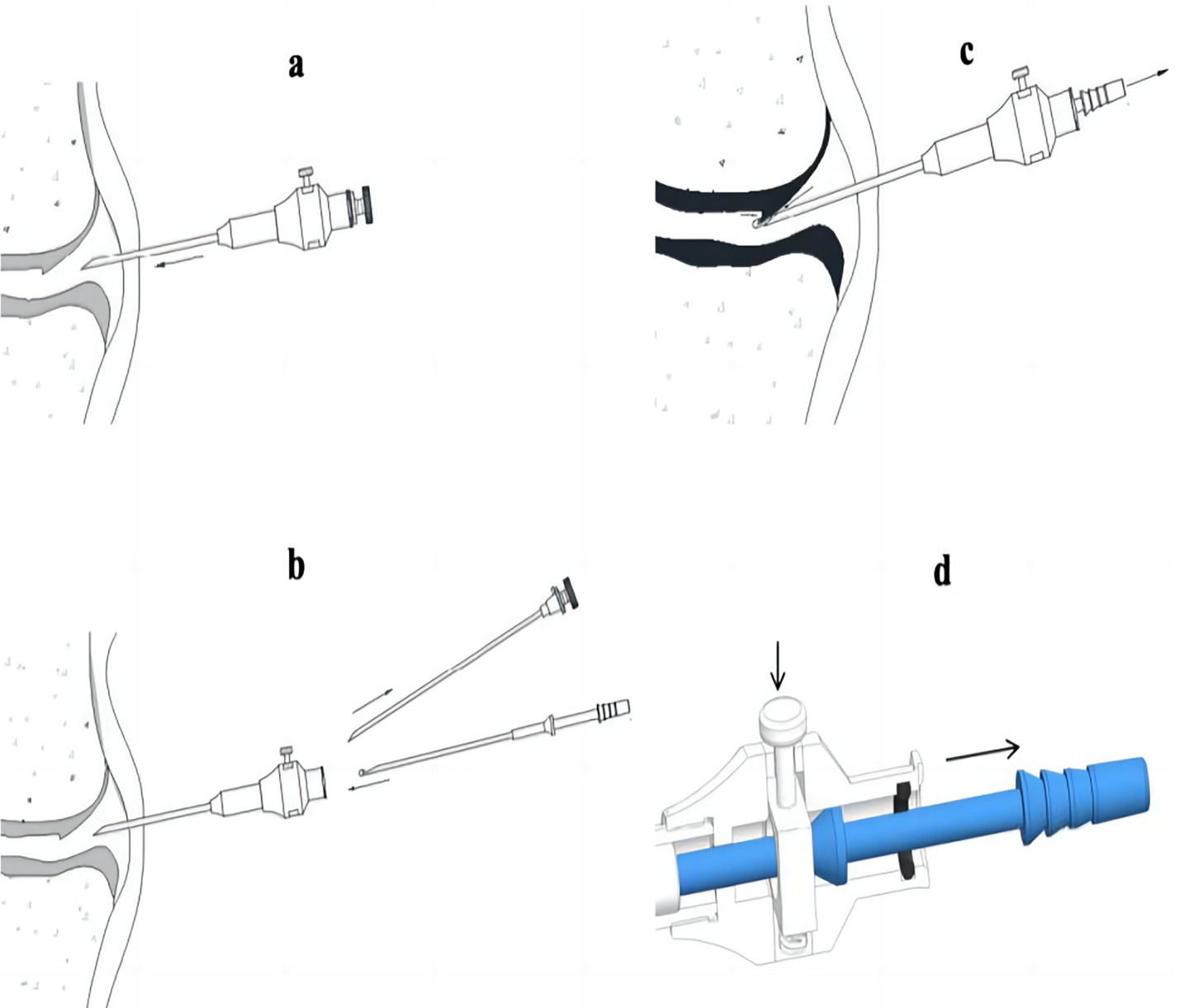

**Figure 2 Procedure of the novel synovial biopsy device for joint tissue sampling.** The device is designed to puncture into the joint cavity using the puncture sheath needle and the puncture inner core (A). After removing the puncture inner core, the sampling needle is inserted, and the firing device is pressed, placing the sampling needle in a ready-to-fire state (B). The tail of the biopsy needle tube is connected to a negative pressure device, which creates a vacuum inside the closed-end hollow section of the biopsy needle tube, ensuring that synovial tissue enters the groove of the sampling needle under the effect of negative pressure (C). Coordinating with the firing device of the biopsy outer sheath, the biopsy needle tube rapidly springs out, and the hooked end at the head of the biopsy needle tube cuts the synovial tissue (D).

was found to be significantly longer in the non-responder group compared to the responder group ($P = 0.008$). The utilization rate of MTX was higher in the non-responder group ($P = 0.026$).

**Table 1 Baseline characteristics of all RA patients in two groups.**

| Characteristic | Responder group ($n$ = 18) | Non-responder group ($n$ = 9) | P |
|---|---|---|---|
| Age (years) | 55.44 ± 9.19 | 46.11 ± 17.97 | 0.17 |
| Female ($n$, %) | 13 (72.22) | 7 (77.78) | 1.00 |
| Disease duration (months) | 57.94 ± 67.18 | 149.33 ± 94.89 | 0.008 |
| X-ray grade | 1.17 ± 0.92 | 1.44 ± 1.13 | 0.23 |
| Rheumatoid factor (IU/ML ) | 245.78 ± 541.48 | 91.53 ± 96.37 | 0.41 |
| ACPA (IU/ML) | 130.27 ± 251.91 | 185.34 ± 324.23 | 0.63 |
| ESR (mm/h) | 40.39 ± 33.76 | 23.56 ± 18.39 | 0.11 |
| CRP (mg/l) | 20.19 ± 23.88 | 23.55 ± 35.78 | 0.77 |
| SJC28 | 2 (2, 6) | 3 (1, 6) | 0.94 |
| TJC28 | 2 (2, 6) | 3 (1, 7.5) | 0.94 |
| VAS | 67.50 (56, 71) | 60 (55, 70) | 0.61 |
| DAS28-CRP | 4.38 ± 0.21 | 4.28 ± 0.39 | 0.80 |
| Combined medication ($n$, %) | | | |
| Glucocorticoid | 15 (83.33) | 6 (66.67) | 0.37 |
| MTX | 10 (55.56) | 9 (100) | 0.026 |
| HCQ | 7 (38.89) | 4 (44.44) | 1.00 |

Note:
Abbreviation: RA, rheumatoid arthritis; RF, rheumatoid factor; ACPA, anti-citrullinated protein antibody; ESR, erythrocyte sedimentation rate; CRP, C reactive protein; SJC28, swollen joint count from 28 joints; TJC28, tender joint count from 28 joints; VAS, visual analog scale for pain assessment; DAS28-CRP, disease activity Score in 28 joints based on C reactive protein; MTX, methotrexate; HCQ, hydroxychloroquine.

## Synovial characteristics of RA patients in two groups

Between the responder and non-responder cohorts, there were no significant variances observed in synovial lining hyperplasia, stromal activity, or inflammatory infiltration. Notably, the non-responder cohort displayed markedly elevated levels of neovascularization compared to responders ($P < 0.05$), as depicted in Figs. 3 and 4. In terms of cell counts for CD3, CD20, CD68, and CD138, a substantial disparity was observed between the two groups. Specifically, the cell counts of CD20 were significantly lower in the responder group in contrast to the non-responder group ($P < 0.05$), as illustrated in Fig. 5.

## Factors associated with poor response to JAKi

Binary logistic regression analysis, after adjusting for age and disease duration, unveiled a notable correlation between inadequate response to JAKi and duration (OR = 1.016, $P = 0.016$), as well as synovial neovascularization (OR = 11.132, $P = 0.016$). Moreover, subsequent multivariable logistic regression analysis reaffirmed synovial neovascularization (OR = 16.157, $P = 0.039$) and disease duration (OR = 1.015, $P = 0.033$) as independent risk factors for poor response to JAKi treatment (refer to Table 2).

## DISCUSSION

RA is a progressive and erosive joint disorder characterized by distinct synovitis, which serves as the primary driver of joint damage in RA. Positioned amidst joints, tendons, and

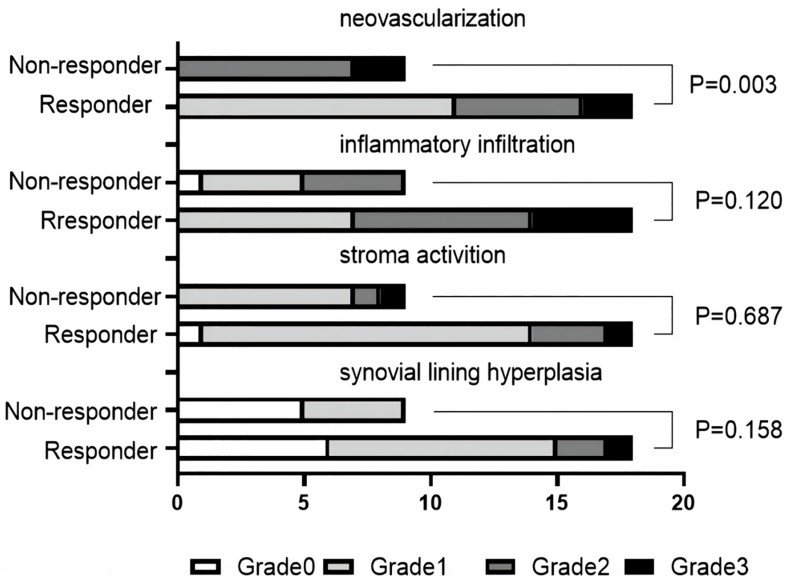

**Figure 3** Comparison of synovial histologic changes (neovascularization, inflammatory infiltration, stroma activation, and synovial lining hyperplasia) in two groups.

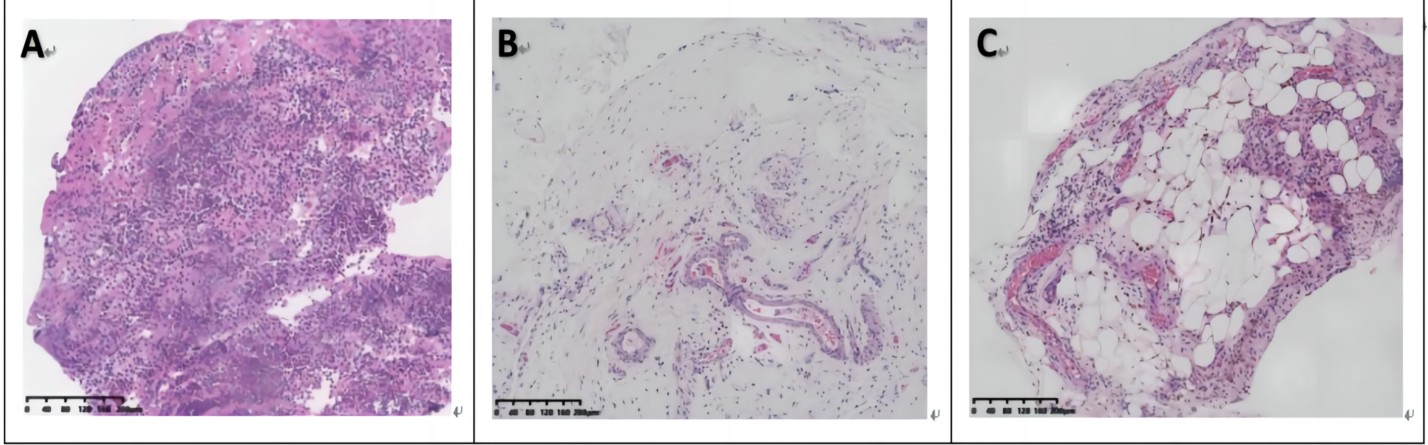

**Figure 4** Synovial histology images of responder (A) and non-responder groups (B and C) (hematoxylin and eosin staining, 10× magnification). (A) Synovial lining hyperplasia with a significant infiltration of immune cells in the sublining. (B and C) Thickening and dilation of small blood vessel walls, as well as congestion with less synovial lining hyperplasia and inflammatory infiltration.

bursae, the synovium plays a pivotal role in maintaining joint health and function. Consisting of two primary layers—the lining layer and the sublining layer, with a mere 5 mm thickness—the synovium undergoes significant structural changes in response to inflammation in RA.

Upon the onset of inflammation, an influx of inflammatory cells infiltrates the sublining layer of the synovium, leading to thickening of the lining layer. Persistent inflammation exacerbates synovial lining hyperplasia, thereby fostering neovascularization within the

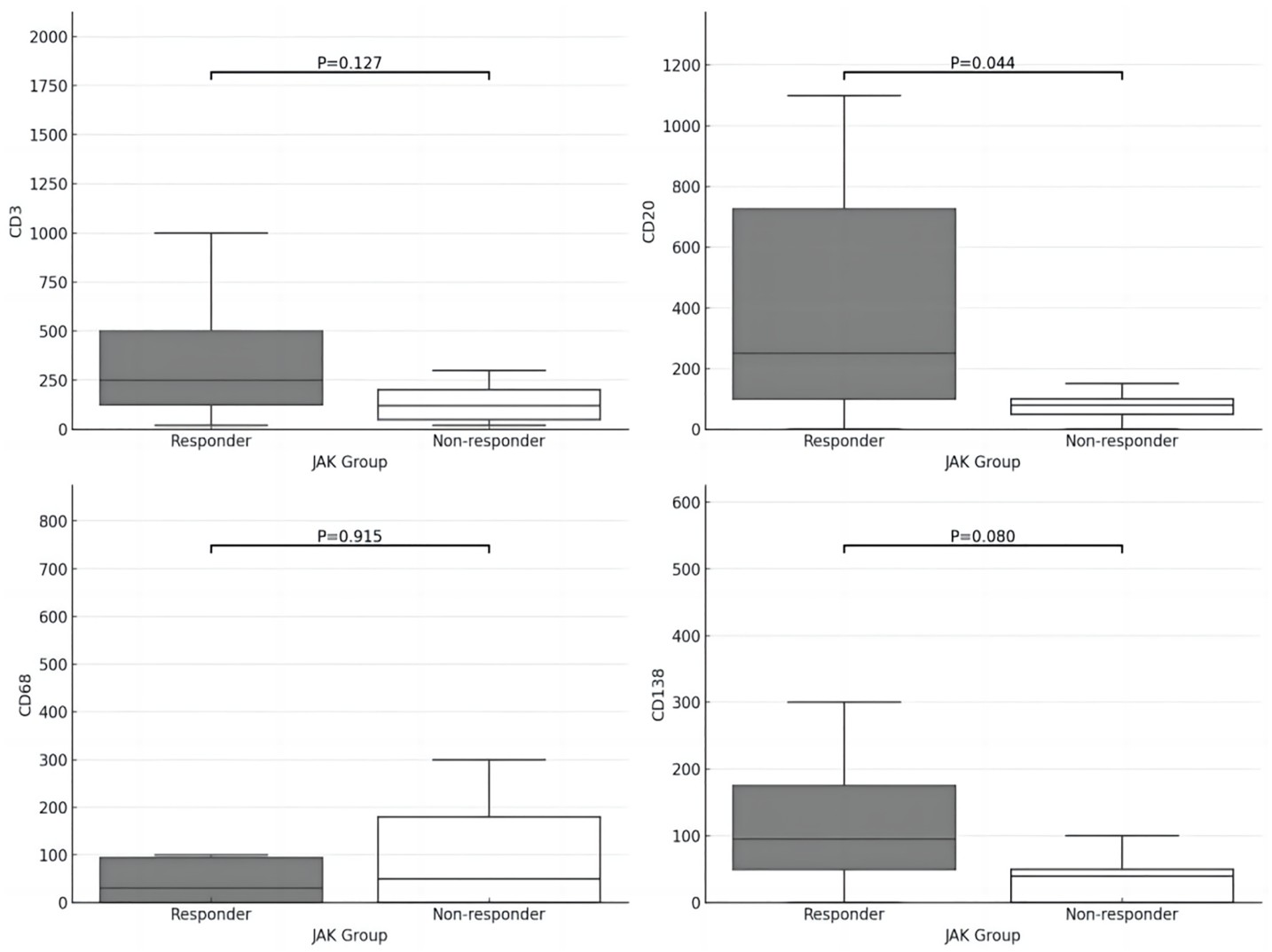

**Figure 5 Comparison of immune cell infiltration (CD3, CD20, CD68, CD138) in synovial sublining between two groups.**

**Table 2 Logistic regression analysis of factors associated with poor response to JAKi.**

| Variables | Model I | | | Model II | | |
|---|---|---|---|---|---|---|
| | OR | 95% CI | *P* | OR | 95% CI | *P* |
| Duration | 1.016 | [1.003–1.029] | 0.016 | 1.015 | [1.001–1.029] | 0.033 |
| Neovascularization | 11.132 | [1.579–78.473] | 0.016 | 16.157 | [1.153–226.371] | 0.039 |

sublining layer. These newly formed blood vessels serve to supply oxygen and nutrients to the proliferating synovial cells, thereby perpetuating synovial proliferation and exacerbating joint damage. This process of neovascularization assumes a critical role in

sustaining the inflammatory cascade and driving joint destruction in RA (*McInnes & Schett, 2011*; *Marrelli et al., 2011*).

The mechanism underlying synovial neovascularization in RA is intricate, governed by a plethora of mediators and cytokines (*Brandao et al., 2023*; *Szekanecz & Koch, 2007*). Among the principal regulatory factors implicated in this process are HIF-a (hypoxia-inducible factor-alpha) and VEGF (vascular endothelial growth factor) (*McInnes & Schett, 2011*; *Leblond, Allanore & Avouac, 2017*). Despite several studies demonstrating the inhibitory effect of JAKi on synovial neovascularization in RA (*Szekanecz et al., 2009*), it is important to note that synovial vascularization represents a chronic and protracted process, often marking the transition from the acute to chronic phase of RA (*Quinonez-Flores, Gonzalez-Chavez & Pacheco-Tena, 2016*). Consequently, targeting this process therapeutically poses considerable challenges.

This study observed that patients exhibiting a poor response to JAKi displayed significantly heightened synovial neovascularization compared to the responder group. Following multivariate analysis, synovial neovascularization persisted as a risk factor for inadequate response to JAKi. This suggests that a synovial vascular phenotype could serve as a predictive indicator for non-response to JAKi.

RA is characterized by its complexity and high heterogeneity, traits that are evident in RA synovial tissue both histologically and molecularly. Recent research has proposed four major phenotypes of RA synovium: lymphoid (dominated by B cells and plasmablasts), myeloid (characterized by macrophages and NF-κB processes), fibroid (comprising hyperplastic but pauci-immune tissues), and low inflammatory (*Dennis et al., 2014*). These synovial phenotypes have been shown to correlate with responses to biologic therapeutics. For instance, RA patients with the myeloid phenotype tend to exhibit the most robust response to anti-TNFα therapy. Additionally, studies have observed an association between the pretreatment expression of genes related to angiogenesis and the clinical response to anti-TNFα, suggesting that the presence of synovial neoangiogenesis may contribute to a favorable outcome following TNFα blockade (*Rivellese et al., 2020*).

Our study compared several histological manifestations of synovial pathology, such as synovial lining hyperplasia, stromal activity, inflammatory infiltration, and neovascularization. We found that RA patients with poor response to JAKi exhibited significantly pronounced vascular lesions in the synovium. Therefore, it is plausible that when prominent neovascularization is observed in the synovium, TNF inhibitors may be a more suitable treatment option.

Our study further validated that disease duration acts as a predictive marker for a diminished response to JAKi. However, employing disease duration to forecast the efficacy of JAKi in the context of early diagnosis and targeted treatment of RA is not practical. The goal of achieving early disease control in RA necessitates indicators that can guide treatment decisions more effectively from the onset. In this regard, our findings suggest that synovial neovascularization could serve as a valuable early predictive marker for guiding clinical treatment decisions. This marker offers a more immediate reflection of

disease activity and therapeutic response potential, thus providing a practical approach for optimizing early treatment strategies in RA.

Interestingly, the levels of observed immune cells (CD20, CD3, CD138) tended to be lower in the non-responder group, though most did not reach statistical significance except for CD20. This is consistent with previous research suggesting that RA patients with a low-inflammatory synovial subset tend to exhibit poor clinical responses to therapy (*Dennis et al., 2014*). However, despite the lower level of synovial immune cell infiltration in the non-responder group, their clinical disease activity score (DAS28-CRP) was not lower.

Research has classified synovial tissue into two phenotypes based on the abundance of synovial B cells: B cell-rich and B cell-poor, with differences in clinical characteristics and treatment responses between the two (*Rivellese et al., 2020*). Our findings demonstrated that RA patients who responded to JAKi therapy had higher levels of synovial CD20+ B cells ($P = 0.044$), suggesting that the B cell-rich phenotype may be associated with a more favorable response to JAKi.

Previous studies have shown that JAK inhibitors exert their anti-inflammatory effects by suppressing the JAK-STAT pathway, thereby inhibiting the signaling of various cytokines. Among these cytokines, IL-4, IL-6, and others have been shown to influence B cell differentiation. Consequently, for RA patients exhibiting a synovial phenotype characterized by B cell-rich infiltration, JAKi therapy may yield particularly promising outcomes.

Our findings underscore the substantial influence of synovial pathophysiological diversity on the clinical response to JAK inhibitor therapy in RA patients. Particularly noteworthy is the observation that patients with a synovial phenotype characterized by neovascularization with low inflammtory infiltration are less responsive to JAK inhibitor treatment. These insights highlight the potential utility of synovial biomarkers in predicting responses to targeted therapies in RA, offering a promising avenue for personalized treatment strategies in this complex autoimmune disease.

Due to the small sample size in this study and the fact that the samples were primarily drawn from a single center, the findings may not be broadly generalizable to other regions or the wider population. Additionally, while the vast majority of patients in this study used Tofacitinib, we included three patients who used Baricitinib in order to adhere to the original study design, which aimed to include all patients using JAK inhibitors. An additional analysis excluding these three patients yielded consistent statistical results, and thus we retained them in accordance with the original registered protocol. Future research with a larger sample size and focused analyses on specific JAK inhibitors is recommended to enhance the generalizability and specificity of the results.

## CONCLUSIONS

In summary, our study revealed that RA patients with poor response to JAKi exhibit significantly increased synovial neovascularization (OR = 16.157, $P = 0.039$) and longer disease duration (OR = 1.015, $P = 0.033$). Increased synovial neovascularization, along

with low inflammation, serves as a potentially valuable early predictive indicator for guiding clinical treatment decisions.

### Funding
The authors received no funding for this work.

### Competing Interests
The authors declare that they have no competing interests.

### Author Contributions
- Mengxia Liu conceived and designed the experiments, performed the experiments, analyzed the data, prepared figures and/or tables, authored or reviewed drafts of the article, and approved the final draft.
- Pengcheng Liu performed the experiments, prepared figures and/or tables, and approved the final draft.
- JianBin Li performed the experiments, analyzed the data, authored or reviewed drafts of the article, and approved the final draft.
- Yiping Huang performed the experiments, prepared figures and/or tables, and approved the final draft.
- Rui Wu conceived and designed the experiments, prepared figures and/or tables, authored or reviewed drafts of the article, and approved the final draft.

### Human Ethics
The following information was supplied relating to ethical approvals (*i.e.*, approving body and any reference numbers):

The ethical review board of the First Affiliated Hospital of Nanchang University granted ethical approval for this study under the ethics number: IIT[2023] Clinical Ethics Review No. 011.

### Data Availability
The raw data are available in the Supplemental File.

### Supplemental Information
Supplemental information for this article can be found online at http://dx.doi.org/10.7717/peerj.18631#supplemental-information.

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
