# Peer review of "Vascular synovial phenotype indicates poor response to JAK inhibitors in rheumatoid arthritis patients: a pilot study"

_PeerJ, doi:10.7717/peerj.18631_

## Round 0.1 · original submission · Major Revisions

The manuscript describes a relevant approach to JAKi in RA as therapy. The authors need to include a sentence that declares the limitations of the study (such as the limited number of samples). The title could be improved by the description of ¨pilot study¨. Clarify the statistical analysis as a statement and remark statistic significance when applied.

Include a point-by-point response to reviewers.

Reviewer 1 ·

Basic reporting

Dear Authors,
We appreciated your manuscript and the effort you made.
A few minor adjustments are to be considered but the work seems to be overall adequate.
There is a section in the manuscript with a few more errors than the others:

In line #149, “disease” is misspelled. Moreover, we suggest to change “were” in “was” as it is not a subordinate clause and it does not require subjunctive.

In line #154, the subject of the sentence is missing as it was inverted in the sentence. Could you rewrite the sentence to make it more approachable?

In line #155 “However, notably: it might be better to use only one of them.

Experimental design

The study is appropriate but it would be useful to underline that the vast majority of the patients used Tofacitinib.
We all know from literature that JAKi are very different; as such, we expect them to have different effects.
It might be useful to point this out in the discussion as well.

Validity of the findings

No comment.

Additional comments

No further comments

Reviewer 2 ·

Basic reporting

A paper by Liu et al. presents valuable data on vascular synovial phenotype in RA as a factor of poor response to JAK inhibitors. Currently, we need more validated data on the usage of JAK inhibitors, thus, the present study delves with an important issue. The paper is clear, the level of language is appropriate. However, the end of the introduction should be enhanced by adding a clear aim of the study.

Experimental design

The study population is pretty low, as RA is one of the most frequently present rheumatic disease. Thus, I believe the current version of the paper present rather a pilot study in this area, which are worth exploring, especially in finding prognostic factors of good response to JAKi.
How did you assess the normality of the data? Perhaps, when comparing non-normal distributed data regarding two groups it would be better to use the Mann-Whitney test.
Also, only 3 RA patients on baricitinib might bias the results. It would be better to present a cohort of patients with comparable % of different JAKi, or present results regarding only one drug.

Validity of the findings

The conclusions should better adhere to the main results of the study. Statistically significant p-values should be presented rounded to the third decimal place (in the case of very small values, p<0.001), and those with a value > 0.05 to the second decimal place.
Table 1. Could you comment on treatment differences (GCS, MTX, HCQ) between analyzed groups?

Additional comments

In some parts of the manuscript a better explanation on focusing on immune cell infiltration in the analysis should be pointed out.

---

## Round 0.2 · Minor Revisions

Dear authors, the manuscript is improved and is almost ready for acceptance. Please realize the suggestion of the reviewer before acceptance.
Follow the indication to resubmit the last version.

Reviewer 2 ·

Basic reporting

The Authors respond to my comments. Please see my suggestions.
Add Q1-Q3 ranges instead of IQR.
Please change gender into sex, in biomedical research it is better to use sex.
found to be 33.3%, please add No. of cases, n =
Please summarize the predictors of poor response to JAKi in RA in conclusions.

Experimental design

NA

Validity of the findings

NA

Additional comments

Great work. It would be valuable to look for longitudinal studies regarding this topic.

---

## Round 0.3 · accepted · Accept

Dear Authors:

The manuscript is accepted based on all the changes applied by the authors following reviewers' suggestions.

Congratulations.